

# Sex ratios in flux: seasonal dynamics and methodological insights in *Rumex* species

Barbara Pawełek[1], Dagmara Kwolek[2] and Grzegorz Góralski[2]

[1] W. Szafer Institute of Botany, Polish Academy of Sciences, Kraków, Poland
[2] Department of Plant Cytology and Embryology, Institute of Botany, Faculty of Biology, Jagiellonian University in Kraków, Kraków, Poland

## ABSTRACT

Dioecy–separate male and female individuals–occurs in less than 10% of angiosperms. Studying dioecy and plant sex chromosomes is key to understanding sex determination, genetic diversity, and ecological interactions, with implications for crop breeding, conservation, and pollination management. Many dioecious plant species deviate from the 1:1 sex ratio expected under Fisher's principle, displaying male- or female-biased populations. The genus *Rumex* (Polygonaceae) is a valuable model for investigating sex-ratio variation and sex chromosome evolution. The dioecious species *Rumex acetosa* and *R. thyrsiflorus* possess heteromorphic sex chromosomes and typically exhibit female-biased sex ratios in natural populations. However, the environmental and developmental drivers of these biases remain largely unresolved. Previous studies on *Rumex* have often relied on single or infrequent population surveys, potentially overlooking the seasonal dynamics of sex ratios, especially given phenological differences between sexes. Female plants remain morphologically recognizable for longer periods due to seed set, which can bias sex-ratio estimates when sampling is limited to specific reproductive stages or time points. To address these limitations, we systematically investigated the temporal dynamics of observable sex ratios in *R. acetosa* and *R. thyrsiflorus* throughout an entire growing season using regular monitoring. Our results demonstrate that observable sex ratios in these populations are dynamic, with significant seasonal fluctuations driven by differences in developmental timing and the duration of morphological recognizability between sexes. As a result, the period during which field-observed sex ratios accurately reflect the true population structure is both limited and highly dependent on the timing of observations. Notably, the earlier flowering of male plants in *R. acetosa* can lead to male-biased sex ratios in early-season surveys, while the prolonged recognizability of fruiting females may bias estimates toward females later in the season.

Corresponding authors
Barbara Pawełek,
b.pawelek@botany.edu.pl
Dagmara Kwolek,
dagmara.kwolek@uj.edu.pl

## INTRODUCTION

Unlike in the animal kingdom, separate male and female individuals are rarely observed in plant species. Dioecy, the presence of distinct unisexual individuals, each producing

either male or female gametes, occurs in only 6% of all angiosperms (*Renner, 2014*), in contrast to gymnosperms, for which scientific reports indicate values reaching up to 65% (*Walas et al., 2018*). Furthermore, advancements in chromosome detection methodologies have led to a significant increase in the number of angiosperm species recognized as possessing sex chromosomes, rising from an initial estimate of approximately 40 (*Ming, Bendahmane & Renner, 2011*) to around 150 species (*Garcia et al., 2023*). Studies on dioecy and various features of sex chromosomes in plants are crucial for understanding various aspects of sex determination, including its evolution, genetic diversity, adaptation, and ecosystem dynamics. They also have practical implications for dioecious crop cultivation, conservation of dioecious species, and pollination management (*Vyskot & Hobza, 2015*; *Carey, Yu & Harkess, 2021*; *Renner & Müller, 2022*; *Kazama, Kobayashi & Filatov, 2023*).

The advancement of DNA and RNA sequencing technologies, particularly Next Generation Sequencing (NGS), has enabled the investigation of sex chromosomes using whole-genome and transcriptome data. This approach permits not only the identification of sequences specific to the X and Y chromosomes, but also the analysis of the functions of sex-linked genes and additional features (*e.g.*, Y chromosome degeneration, detection of introgression signals, transposable elements, or comparisons between 'old' and 'new' sex-linked regions) associated with these chromosomes (*Sacchi et al., 2024*; *Hibbins et al., 2025*). Such research has been performed, among others, in kiwifruit (*Actinidia*) (*Akagi et al., 2018*) and sorrel (*Rumex*) (*Michalovova et al., 2015*; *Jesionek et al., 2021*).

One of the most intriguing questions in the study of dioecy is why biased sex ratios are observed in many dioecious species within natural populations. According to Fisher's principle of classical evolutionary theory, populations are expected to maintain a 1:1 sex ratio, assuming equal parental investment in male and female offspring and similar environmental effects on both sexes (*Fisher, 1930*). However, the occurrence of sex-biased populations in dioecious plants challenges this theoretical expectation. Understanding the factors driving male- or female-bias provides valuable insights into the selective pressures and ecological dynamics shaping sex determination.

Recent analysis of angiosperm families revealed that approximately half of dioecious species show significant deviations from parity, with male bias being nearly twice as common as female bias (*Field, Pickup & Barrett, 2013*). Male-biased populations are observed more frequently in stressful environments, whereas female-biased populations are more common in resource-rich or stable conditions and among species with heteromorphic sex chromosomes. These biases are thought to result from sex-specific resource allocation, with females incurring higher reproductive costs due to seed production, leading to higher mortality risks in resource-limited environments (*Lloyd, 1974*; *Korpelainen, 1998*; *Munné-Bosch, 2015*; *Lei et al., 2017*).

Several factors have been proposed to influence sex ratios in natural plant populations, encompassing both prezygotic and postzygotic mechanisms that disrupt sex ratios at various stages of individual development. Prezygotic mechanisms include maternal tissue support of pollen tube growth, certation (*Correns, 1917*; *Correns, 1922*; *Correns, 1928*), and sex ratio distorter/restorer systems (*Taylor, 1994*).

Postzygotic mechanisms, which are considered highly influential in altering sex ratios, include zygotic drive (*Rice, Gavrilets & Friberg, 2008*), selective embryo abortion by the maternal plant (*Varga & Soulsbury, 2020*), sex-specific mortality, and environmental effects on sex expression. In some dioecious plants, males and females exhibit differential survival rates due to genetic or environmental factors. For instance, females may experience higher mortality in resource-limited environments due to the greater reproductive costs associated with seed production, whereas males may be more vulnerable under other conditions (*Stehlik, Friedman & Barrett, 2008*; *Barrett et al., 2010*; *Liu, Korpelainen & Li, 2021*; *Bürli, Pannell & Tonnabel, 2022*). For example, in *Rumex nivalis*, female-biased sex ratios deepen postzygotically due to higher mortality rates among male offspring during the following developmental stages (*Stehlik & Barrett, 2005*).

Environmental factors such as temperature, nutrient availability, and stress are considered to play a crucial role in establishing sex ratios in natural populations (*Putwain & Harper, 1972*; *Weiner, 1990*; *Korpelainen, 1991*; *Li et al., 2019*). Although environmental factors are thought to have less influence on female frequency in dioecious systems compared to subdioecious and gynodioecious systems (*Retuerto, Vilas & Varga, 2018*), they can still lead to sex-biased mortality or shifts in sex ratios during development. For instance, in *R. nivalis*, the distance between male and female individuals and the pollen load can affect the sex ratio of seeds, with higher pollen loads favoring female-biased progeny due to certation (*Stehlik, Friedman & Barrett, 2008*). Ecological and environmental factors can also impact pollen and seed production (*Dorken & Pannell, 2008*) and sex-dependent mortality associated with the costs of maintaining each sexual function (*Obeso, 2002*).

The genus *Rumex* comprises approximately 200 species that are distributed worldwide, on most continents (Europe, Asia, Africa, and America). They are used for various purposes, including in local cuisines and for medical purposes in many parts of the world (*Vasas, Orban-Gyapai & Hohmann, 2015*). It is also a useful model for studying sex chromosomes, sex-determination systems, and sex ratios in populations. Because it includes monoecious, dioecious, gynodioecious, and hermaphroditic species, it enables research on the evolution of sexual dimorphism in plants. Some dioecious species have heteromorphic sex chromosomes. Some of them have an XX/XY chromosome system, but there are also those with an $XX/XY_1Y_2$ system, including *R. acetosa* and *R. thyrsiflorus* (*Navajas-Pérez et al. 2005a*; *Grabowska-Joachimiak et al., 2015*).

Despite the general trend, female-biased sex ratios are observed in natural populations of *R. acetosa* L. (common sorrel) and *R. thyrsiflorus* Fingerh. (thyrse sorrel), both members of the family Polygonaceae, subgenus *Acetosa* (*Putwain & Harper, 1972*; *Zarzycki & Rychlewski, 1972*; *Rychlewski & Zarzycki, 1981*; *Rychlewski & Zarzycki, 1986*; *Korpelainen, 1992*; *Korpelainen, 2002*; *Błocka-Wandas et al., 2007*). Sex ratio deviations in these species occur at various developmental stages, including pollen grains and seeds (*Błocka-Wandas et al., 2007*; *Kwolek & Joachimiak, 2011*; *Bizan et al., 2014*). These species serve as model organisms for studying sex chromosomes and sex ratio biases in natural populations. Both are common, widely distributed perennial dioecious herbs found in meadows, grasslands, pastures, wastelands, and roadsides. *R. acetosa* typically grows in higher, seldom-flooded zones with clayish substrates, while *R. thyrsiflorus* thrives in sandy soils in

similar riverine environments (*Świetlińska, 1963*; *Świetlińska, Łotocka-Jakubowska & Żuk, 1970*; *Van Assche, Van Nerum & Darius, 2002*).

Individuals of both species are characterized by heteromorphic sex chromosomes: $2n = 14$ ($6A + XX$) in females and $2n = 15$ ($6A + XY_1Y_2$) in males (*Kihara & Ono, 1925*; *Świetlińska, 1963*; *Zuk, 1970b*; *Navajas-Pérez et al. 2005a*). Sex determination in *R. acetosa* and *R. thyrsiflorus* is not controlled by an active Y chromosome but rather by the X:A ratio (the ratio of X chromosomes to autosome sets) (*Putwain & Harper, 1972*). Individuals with an X:A ratio of $\leq 0.5$ develop as males, while those with an X:A ratio of $\geq 1.0$ develop as females. Intermediate X:A ratios, such as in polyploids and aneuploids, result in intersex or hermaphrodite flowers (*Parker, 1990*; *Parker & Clark, 1991*; *Ainsworth et al., 2005*).

Many studies investigating sex ratios in natural populations of *Rumex* have been based on field observations conducted once or twice during the growing season (in some cases, the frequency of observations is unspecified but can often be assumed to be limited to a single observation per season) and are usually based on visible sex characteristics: flowers and fruits (*Sprecher, 1913*; *Correns, 1922*; *Turesson, 1925*; *Świetlińska, 1963*; *Harris, 1968*; *Zuk, 1970a*; *Putwain & Harper, 1972*; *Zarzycki & Rychlewski, 1972*; *Conn & Blum, 1981*; *Korpelainen, 1991*; *Korpelainen, 1992*; *Korpelainen, 2002*; *Stehlik & Barrett, 2005*; *Pickup & Barrett, 2013*; *Bürli, Pannell & Tonnabel, 2022*). Reported data on the proportion of mature male and female individuals in natural and cultivated populations of *R. acetosa* and *R. thyrsiflorus* are highly inconsistent. Frequencies of females range from 52% to 91.67% (in extreme cases, even 100%) for *R. acetosa* and from 62% to 92.5% for *R. thyrsiflorus*. In some artificially sown populations of *R. acetosa*, no female bias was observed, with male individuals predominating, though the deviation was not statistically significant (*Turesson, 1925*; *Świetlińska, 1963*; *Putwain & Harper, 1972*; *Zarzycki & Rychlewski, 1972*; *Zuk, 1963*; *Korpelainen, 2002*).

In seeds, the male-to-female ratio is closer to 1:1, with male individuals predominating sometimes, though this is often not statistically significant. For *R. acetosa*, the percentage of female seeds ranges from 51% to 59.7%, while for *R. thyrsiflorus*, it ranges from 42.3% to 61.8% (*Zarzycki & Rychlewski, 1972*; *Rychlewski & Zarzycki, 1981*; *Rychlewski & Zarzycki, 1986*; *Korpelainen, 2002*; *Kwolek & Joachimiak, 2011*).

However, the above-mentioned approaches may not fully capture the dynamics of sex ratios in populations. Differences between individuals of the two sexes in *Rumex,* as in the case of most plants, are practically limited to organs involved in sexual reproduction, at least if we consider observations based on their morphology (*Stehlik & Barrett, 2005*). Consequently, differences in flowering time and lifespan between sexes, influenced by environmental conditions and seasonal timing, can significantly affect observed sex ratios, potentially leading to incomplete or misleading conclusions. For example, males are not recognizable after flowering, while fruit-bearing females remain visible for longer periods (from the start of flowering at least to the end of the fruit's presence) (*Putwain & Harper, 1972*; *Korpelainen, 1998*). These obstacles can significantly influence the results depending on the time of observation; the same population can give different outcomes when studied even at different points in the same season. Therefore, it is important to develop the proper methodology of observations. To ensure robust results, the methodology should

be optimized for the rate and nature of changes observed in the studied plant species and remain consistent throughout the study duration. To achieve this goal, it is indispensable to analyze the changes of sex recognizability during the year systematically, especially with the relationship to the real sex proportions in the population.

In this study, we aimed to investigate how sex ratio observation results change throughout the growing season. By performing regular analyses of the number of *R. acetosa* and *R. thyrsiflorus* individuals, we demonstrate the importance of systematic field studies over an entire season to assess sex ratio disturbances in natural populations accurately.

## MATERIALS & METHODS

### Field analysis

Seeds used for sex determination and sowing were randomly selected from the obtained seed pools. Seeds of *R. acetosa* were collected in 2016 in the field located in Kraków, Lesser Poland Voivodeship, Poland (50°01′31.65″N, 19°53′39.37″E) from plants growing in natural populations. In the case of *R. thyrsiflorus*, seeds were purchased commercially (PlantiCo, Krakow, Poland). These seeds were offered as *R. acetosa*, but molecular analyses, based on a molecular marker (see below), proved that they belong to the *R. thyrsiflorus* species. Seeds from these pools were later used for seed sex determination (Fig. S1), and the rest are stored in the collection of the Botanic Garden of Jagiellonian University in Kraków. The use of commercial seeds requires explanation. Some researchers have observed crosses between the species studied (Świetlińska, 1963; Świetlińska, Łotocka-Jakubowska & Żuk, 1970); however, the probability of obtaining pure seeds is much higher for *R. acetosa* than for *R. thyrsiflorus*. This occurs because the former flowers earlier, reducing the likelihood of cross-pollination by other species. Conversely, when the latter flowers, some *R. acetosa* plants may still produce pollen grains and fertilize *R. thyrsiflorus* females, producing hybrids. Therefore, using commercially available seeds from a specific line would likely resolve this issue.

The study field was established at the Institute of Botany, Jagiellonian University in Kraków, Poland (50°01′38.7″N, 19°54′12.2″E) under semi-controlled conditions. The field was partially shaded by building walls, with maximum sunlight exposure at midday and the length of light exposure depending on the season. The soil was initially prepared by digging and weeding the soil to ensure the best growing conditions. Additionally, plants were watered once or twice a day during the hottest period. Temperature data were obtained from the climatology station (located 135 m from the experimental field) of the Department of Climatology, Institute of Geography and Spatial Management, Jagiellonian University.

On May 18, 2020, 104 seeds of each species were sown into commercially obtained peat discs, which typically have a pH of approximately 6. Upon seedling emergence on August 2, all individuals of *R. acetosa* and *R. thyrsiflorus* were transferred to plant pots. After approximately two months, on October 11, all seedlings that germinated and survived—35 individuals of *R. acetosa* and 26 individuals of *R. thyrsiflorus*—were transplanted into the soil at a spacing of 15–20 cm. By this time, nearly all plants had developed leaves

measuring 150 mm or more. During cultivation, the plants were irrigated; no fertilization was applied. It is worth noting that these species occur in semi-natural and artificial communities, such as mown meadows and pastures. To simulate mowing, the plants were cut to approximately 10 cm above ground on June 23 (Fig. S2). The developmental dynamics of *R. acetosa* and *R. thyrsiflorus* were monitored throughout the entire growing season of 2021. Key phenological stages were analyzed, including post-winter regrowth, the emergence of male and female flowers, the pollination phase in male individuals, seed production and maintenance in female individuals, and the senescing stage preceding winter dormancy. The sex of mature plants was determined by the presence of male or female flowers, with stamens or pistils, respectively (Fig. S3). In the case of seeds and one specimen that did not produce flowers during the experiment, molecular methods were used (see below). To assess the sex ratio in seeds and mature plants of the two species, we calculated the frequency of males as the proportion of male specimens out of the total. This was done for both seeds and sexually mature individuals—identified by male or female flowers, the presence of seeds, or, in one case, molecular markers.

Developmental stages were defined as follows: leaves (plants without flowers), flowering (closed flower buds and open flowers), pollination phase—for male plants (starts with the first flower to pollinate), seed production phase—for female plants (starts with the first flower with seed), and senescing (plants turning yellowish after the pollination phase or with seeds).

## Molecular analysis

Molecular sex determination of seeds and one plant was performed by procedures based on the method described by *Kwolek & Joachimiak (2011)*. Briefly, genomic DNA was extracted from a single seed or leaf using a CTAB-based protocol. PCR amplification was carried out with two primer pairs that target male-linked DNA markers on the Y chromosome, thereby enabling the identification of male individuals: (1) RAY-f and RAY-r (*Korpelainen, 2002*); (2) URG08-F and URG08-R (*Mariotti et al., 2009*). Additionally, the R730-A and R730-B (*Navajas-Pérez et al., 2005b*) primers were used to verify the quality of the extracted DNA. The PCR products were separated by agarose gel electrophoresis. The presence or absence of male-specific bands enabled the determination of the sex of each specimen. To confirm the correct identification of male seed species, the differences between PCR products obtained with primers URG08-F and URG08-R in male individuals of *R. acetosa* and *R. thyrsiflorus* were applied - while in *R. acetosa*, PCR amplification with URG08-F and URG08-R yields a single product of approximately 700 bp, in *R. thyrsiflorus* PCR amplification with the same primers yields two products: one of approximately 700 bp (the same size as in *R. acetosa*) and a shorter fragment of about 600 bp. The shorter DNA fragment (600 bp) in *R. thyrsiflorus* is the result of a large indel of 110 bp compared to the 700 bp product (Figs. S1, S4) (*Kwolek & Joachimiak, 2011*; *Grabowska-Joachimiak et al., 2012*).

## Statistical analysis and digital processing

All statistical analyses and graphical representations were conducted in the R programming environment using RStudio v4.4.3 (*R Core Team, 2025*). Graphical adjustments were

**Table 1  Sex proportions in seeds and mature plants in *R. acetosa* and *R. thyrsiflorus*.** M, male number; F, female number; M freq, frequency of males; *p*, *p* value calculated by exact binomial, asterisks indicate statistical importance respecting Benjamini & Hochberg correction (*Benjamini & Hochberg, 1995*).

| Species | Stage | M | F | M freq | *p* |
|---|---|---|---|---|---|
| *R. acetosa* | Seeds | 46 | 54 | 0.46 | 0.4841 |
| *R. acetosa* | Mature | 17 | 18 | 0.49 | 1.0000 |
| *R. thyrsiflorus* | Seeds | 80 | 157 | 0.34 | *0.0000 |
| *R. thyrsiflorus* | Mature | 9 | 17 | 0.35 | 0.1686 |

performed using Inkscape v1.3.2 or v1.4 (*Jeanmougin & Hachmann, 2023*) when necessary. The proportion of males was calculated as the number of males divided by the total number of seeds or plants counted. To test whether the proportion of males in seeds and mature plants differed significantly from 0.5, an exact binomial test was used (*via* the *binom.test()* function). The visibility of sex characteristics (flowers, pollen, fruits) was analyzed using the Student's *t*-test (*via* the *t.test()* function). Statistical significance was determined if the *p*-value was lower than the critical values calculated for $\alpha = 0.05$ using the Benjamini & Hochberg correction (*Benjamini & Hochberg, 1995*) for a given set of comparisons. Where the results were intended to simulate one-point observations at different times of the growing season, each observation was treated independently, and $p < 0.05$ was used as a critical value without correction.

# RESULTS

## Seed germination

Individuals of *R. thyrsiflorus* began to germinate earlier than *R. acetosa*, and the young seedlings of *R. thyrsiflorus* also reached 150 mm of leaf size earlier. The first two germinating seeds for *R. thyrsiflorus* were observed on May 22, 2020, and the last germinated on June 2, 2020. The first germinating seed of *R. acetosa* was observed on May 25, 2020 (at this time six seedlings of *R. thyrsiflorus* were already visible), and the last on June 9, 2020. In total 208 seeds were sown (104 of each species), 86 germinated (52 of *R. acetosa* and 34 of *R. thyrsiflorus*), and 61 survived (35 and 26 respectively).

## Sex ratio in seeds and mature plants within and between the studied species

Despite the relatively low germination and early survival rates (35 out of 104 individuals in *R. acetosa* and 26 out of 104 individuals in *R. thyrsiflorus*), all surviving individuals planted in the soil withstood the winter and resumed growth in 2021. All individuals flowered, except for one female plant that showed only vegetative growth.

For both species studied, sex ratios, expressed as male sex proportions in seeds and mature plants (individuals with flowers and/or seeds), were calculated. The results are presented in Table 1 and Fig. 1.

The frequency of male seeds was lower than 0.5 for both species, but the deviation was statistically significant ($p < 0.05$) only in *R. thyrsiflorus* seeds. For mature plants, the sex bias was not statistically significant for either species. Moreover, in *R. acetosa*, the numbers

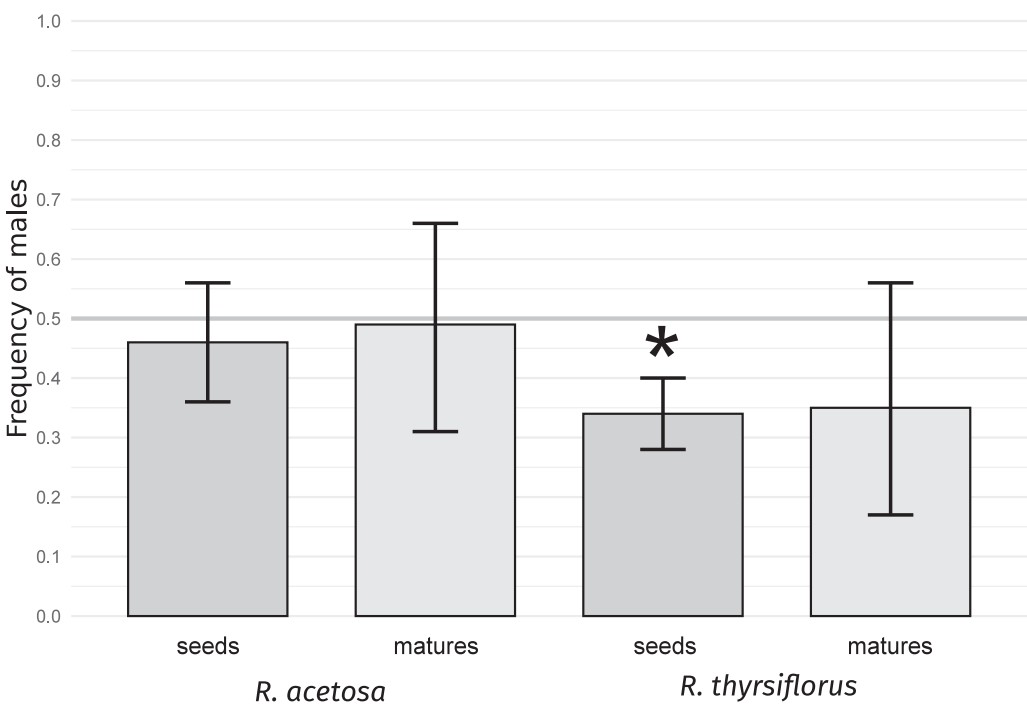

**Figure 1  Frequency of males in seeds and mature plants for *R. acetosa* and *R. thyrsiflorus*.** Asterisks (*) indicate statistically significant differences from a 0.5 proportion, calculated using the exact binomial test with the Benjamini & Hochberg correction (*Benjamini & Hochberg, 1995*). Error bars represent the lower and upper confidence interval values for the proportions. A female bias is observable in both species, with the effect being stronger in the latter; however, it is statistically significant only for the seeds.

of male and female mature plants were nearly equal (17 and 18, respectively). The difference between the frequency of male seeds and mature plants appeared to be greater in *R. acetosa* (0.46 *vs.* 0.49) than in *R. thyrsiflorus* (0.34 *vs.* 0.35); however, these differences were not statistically significant for either species. These results confirm that seeds are biased in favor of the female sex in *R. thyrsiflorus* but this bias, if it exists, is not evident in *R. acetosa* and in sexually mature plants of both species, at least in a relatively small population.

## The development and visibility of sex characteristics in plants

The seasonal variation and duration of visibility of sexual characteristics among species, sexes, and individual plants are illustrated in Figs. 2 and 3, which are described below separately for each species. Figure 2 also includes temperature data from the experiment period, which may serve as background for the plotted phenological changes.

### Rumex acetosa

At the beginning of the 2021 growing season, on April 2, all individuals except one female had already grown shoots with leaves and continued to develop. Almost all male plants (14 from 17) began flowering earlier than females (Fig. 2). The first male individuals with flower buds appeared on April 28, while female flower buds began to emerge five days later,

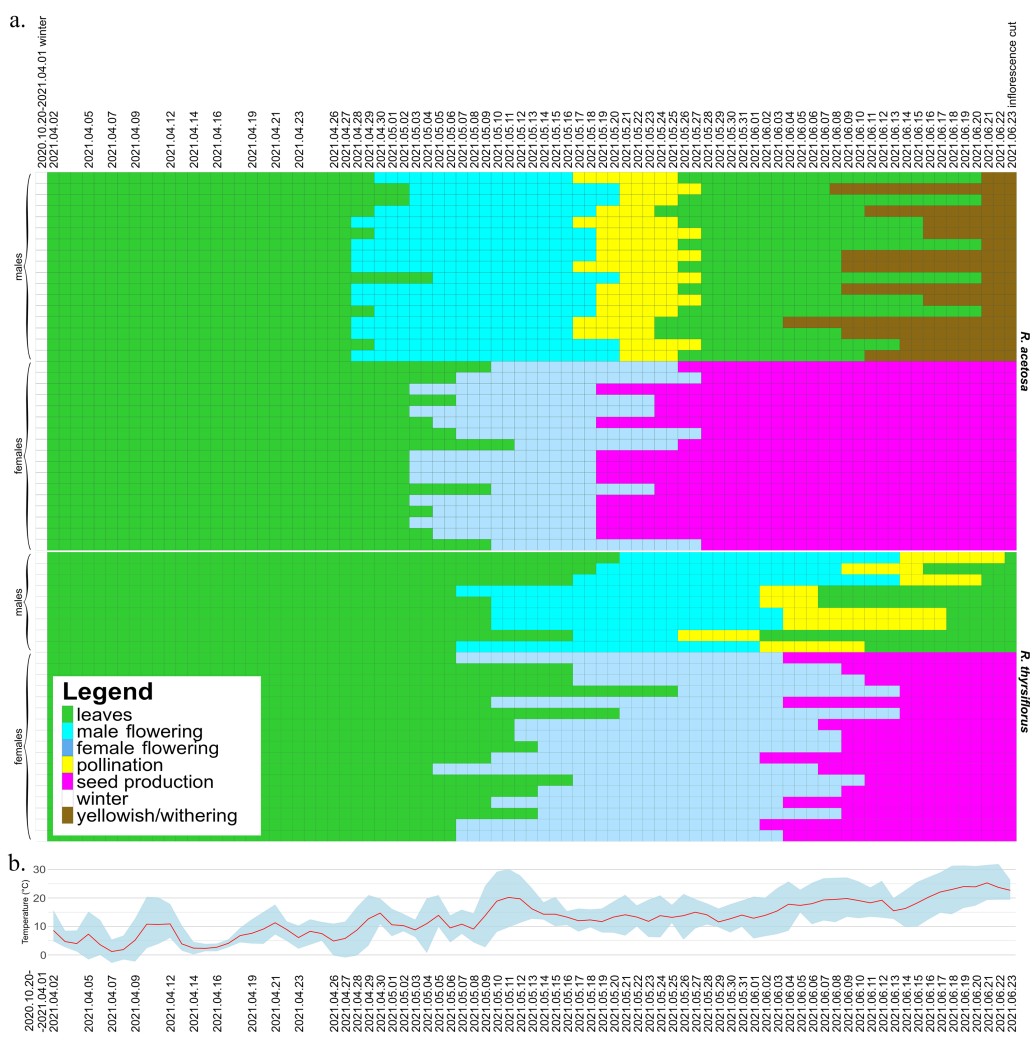

**Figure 2** **Development of male and female individuals of *R. acetosa* and *R. thyrsiflorus* throughout the 2021 growing season, concluding with the cutting of plants, with temperature record.** (A) Each row represents the duration of the developmental stages of a single plant, starting from leaf development after winter dormancy, through flowering, pollination/seed production, and ending with senescence or cutting. The columns represent consecutive days. Observations were conducted at varying intervals depending on the rate of morphological changes occurring in maturing plants. The dates marked on the graph indicate the times when actual measurements were taken; the remainder of the graph, for better visualization of changes over time, is interpolated. One (non-flowering) individual was excluded from the graphic. Visible differences between species, sexes, and individuals are observed, particularly in features such as the timing of flowering onset and the duration of developmental phases. Generally, *R. acetosa* began reproduction earlier, with male flowers appearing earlier than female flowers in this species. (B) Daily minimum and maximum temperatures (grey-blue band) and daily mean temperatures (red line) recorded during the experimental period from April to June 2021. The dates marked on the graph indicate the times when actual measurements were taken.

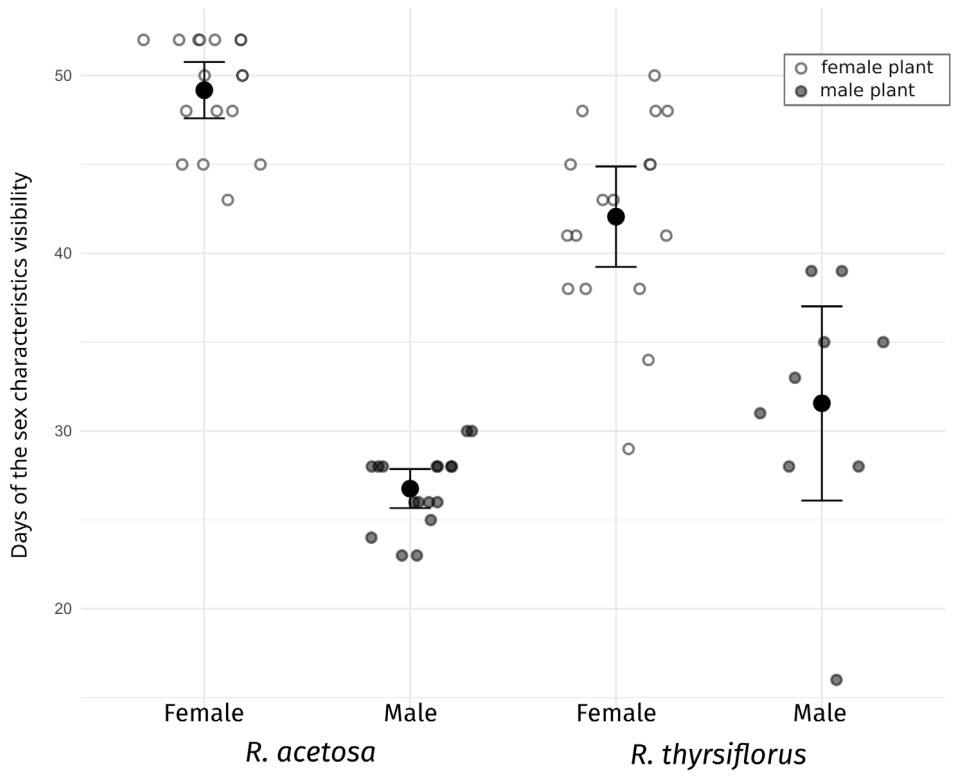

**Figure 3 Visibility of sex characteristics in mature male and female plants of *R. acetosa* and *R. thyrsiflorus*.** Smaller white and grey dots represent data for individual plants (female and male, respectively), larger black dots indicate the mean for each group, and error bars represent the lower and upper confidence interval values for the means. Visible differences in the duration of sex characteristics are observed between plants, sexes, and species; however, the difference between males of the two species is not statistically significant. Generally, sex characteristics were observable for a longer duration in females (flowering and fruiting with seeds) than in males (flowering and pollination phase). See also Table 3.

on May 3. The latest flowering male appeared on May 5, while the latest female flower buds emerged on May 12.

The pollination phase began with the first pollinating male on May 17 and ended with the last plant finishing pollen production on May 27. This phase lasted 11 days, with a single male pollinating for a maximum of 9 days. Females began producing fruits with seeds on May 19. The seed stage lasted for a maximum of 36 days, with an average of 32.7 days, after which all individuals were cut down on June 23 (on this time all females possessed fruits with seeds). Males began to senesce as early as 11 days after being cut, while females remained green for a longer period, starting to turn yellowish 78 days after the cut. Observations concluded on October 29, when all individuals had senesced.

The duration of sex characteristic visibility differed significantly between males and females. Male individuals were visible for a minimum of 23 days and a maximum of 30 days (flowering and the pollination phase), with a mean duration of 26.76 days. In comparison, female individuals were visible for a minimum of 43 days and a maximum

of 52 days (flowering and fruits with seeds), with a mean duration of 49.18 days, and the difference was statistically significant. In summary, female individuals remained distinguishable for nearly twice as long (1.84 times longer) as male individuals during the vegetative season.

At the beginning of the season, due to the earlier emergence of male flowers, the proportion of male individuals in the population was equal to 1. As female flowers began to appear, the sex ratio in the population gradually shifted, initially approaching or reaching a 1:1 ratio (after 13 days) and subsequently becoming skewed in favor of female individuals from day 29 until the cutoff of all plants. Analysis of the temporal changes in the proportion of male individuals shows that the phases during which the male share was approximately or exactly 0.5 and 0.0 were prolonged (10th May to 21th May and since 28th May), whereas the actual transitions in sex ratio occurred rather abruptly (30th April to 5th May and 24th May to 28th May). The proportion of male individuals was the highest (1.0) at the beginning of the growing season (28th April to 30th April) and gradually declined to 0.0 on 28th May. At no point thereafter was an increase in the proportion of male individuals observed (Fig. 4A).

### Rumex thyrsiflorus

At the beginning of the 2021 growing season, on April 2, all plants possessed leaves and continued to develop. Individuals of both sexes began flowering at approximately the same time-the first female plant began flowering on May 5, while the first male flowers were observed on two plants two days later. The latest male started to develop flowers on May 21, while the latest female flowers began to emerge on May 26.

The pollination phase lasted 28 days in total, from May 26 to June 22, with single males producing pollen for the longest of 14 days. Fruits with seeds started on June 2 and lasted until June 23, as on that day all plants were cut down. In contrast to *R. acetosa*, in the *R. thyrsiflorus* population, the seed stage was not the longest developmental stage, with a maximum duration of 22 days and an average of 16.41 days. The most extended stage overall was flowering and pollen production in males, lasting on average 31.56 days. Female flowers started to develop generally later than those of *R. acetosa*, and for this reason, they had less time to exist before being cut down.

The duration of sex characteristic visibility differed significantly between males and females. Male individuals were visible for a minimum of 16 days and a maximum of 39 days (flowering and the pollination phase), with a mean duration of 31.56 days. In comparison, female individuals were visible for a minimum of 29 days and a maximum of 50 days (flowering and fruits with seeds), with a mean duration of 42.06 days, and the difference was statistically significant. In summary, during the vegetative season, female individuals remained distinguishable, on average, for 10.5 days longer than male individuals.

One male and six female individuals flowered again during the 2021 growing season. The second male flowering and pollination period lasted 26 days, from July 23 to August 17. The second female flowers began to appear on July 12. The female flowering and seed-development stage was the longest, lasting 50 days—from July 15 to September 2.

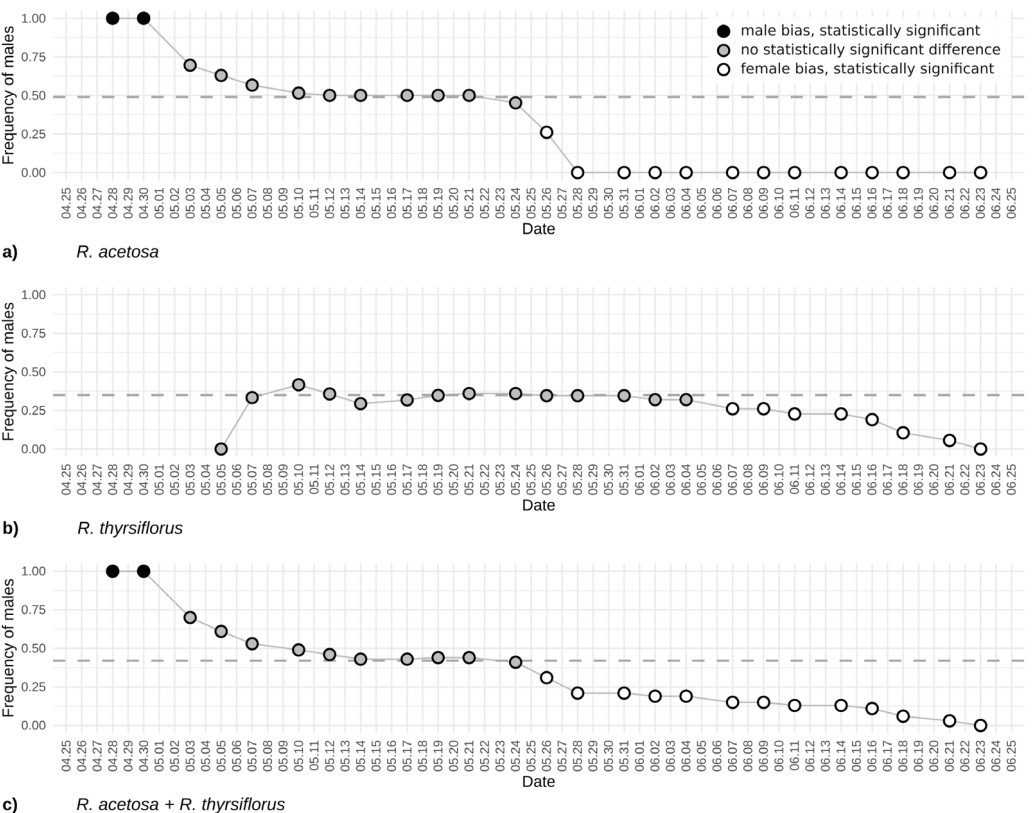

**Figure 4** **The frequency of males based on visible sex features on each observation day during the first flowering period for *R. acetosa* (A), *R. thyrsiflorus* (B), and for both species combined (C).** Black dots indicate male bias that is statistically significant ($p < 0.05$) or where the proportion equals 1; gray dots indicate statistically non-significant results; white dots indicate statistically significant female bias. The dashed line indicates the actual proportion in the population. Observable changes in sex proportion over time are evident. In all cases, female bias is statistically significant only in the later part of the season. At the beginning of the reproductive phase, *R. acetosa* displays male bias due to the earlier onset of flowering.

The first seeds from the second flowering were observed on July 23, and the last seeds were recorded as late as September 7.

What is worth noting is that all individuals that flowered for the second time started to senesce immediately after losing seeds or finishing pollination. In the case of plants that did not repeat flowering, after the pollination phase or cut down, plants stayed green and overall started to turn yellowish later than individuals that repeated flowering. The first plants that flowered once started to senesce on September 8, while individuals that flowered twice began to turn yellow as early as August 9.

When the durations of the first and second flowering periods were combined, male plants were observed for an average of 34.33 days, which is approximately three days longer, while female plants were visible for an average of 55.94 days, exceeding this by over ten days. Furthermore, it is important to acknowledge that observations conducted at the end of the vegetative season may produce data that significantly deviate from the actual number of male and female individuals present within the studied population. Yet, due to

**Table 2** **The visibility of sex characteristics in male and female mature plants of *R. acetosa* and *R. thyrsiflorus* for first flowering.** M, male; F, female; min, max, mean, minimal, maximal, and mean numbers of days of sex characteristics visibility; sd, standard deviation for mean, CI-, CI+, lower and upper values of confidence intervals for means.

| Species | Sex | Min | Max | Mean | sd | CI- | CI+ |
|---------|-----|-----|-----|------|-----|-----|-----|
| *R. acetosa* | M | 23 | 30 | 26.76 | 2.14 | 25.67 | 27.86 |
| *R. acetosa* | F | 43 | 52 | 49.18 | 3.09 | 47.59 | 50.76 |
| *R. thyrsiflorus* | M | 16 | 39 | 31.56 | 7.11 | 26.09 | 37.02 |
| *R. thyrsiflorus* | F | 29 | 50 | 42.06 | 5.51 | 39.23 | 44.89 |

the limited number of individuals flowering for the second time, they were omitted from further analysis.

During the season, female individuals (flowering and with seeds) outnumbered recognizable males at every point. On the day when individuals were cut off, only female individuals were recognizable, as males lost their flowers after the pollination phase. By comparison, the actual proportion of females was 0.65. Individuals of both sexes started the process of dormancy at approximately the same time-the first senesced male individual, the very same that flowered twice, was observed on August 18, and the first female individual to senesce was seen on August 9. In contrast to previous species, the temporal dynamics of male individual frequency in the population of *R. thyrsiflorus* exhibit a different pattern.

In summary, at the beginning of the season, only female individuals were recognizable. Over time, an increasing number of male plants with flowers emerged, gradually raising their proportion in the population and reaching a maximum value of 0.42 on 10th May. The proportion of male individuals fluctuates throughout the season, ultimately declining gradually to 0 prior to the cutoff of all plants (Fig. 4B).

A trend was observed in which individuals that became ochre and senesced earliest were the ones that flowered twice during the growing season. Of the nine earliest senescing individuals, seven were the ones that flowered again. The observations ended on October 29, once all individuals were senesced.

## Durations of sex characteristics' visibility

The visibility of sex characteristics for both species studied is presented in Table 2 and Fig. 3, and statistical analyses between sexes and species are shown in Table 3. In the case of *R. acetosa* the mean time when male plants were identifiable (time of flowering) was about 27 days and females could be recognized (flowering and fruiting) in average c.a. 49 days, however this period was shortened by cutting the plants, and without intervention could be even longer. For *R. thyrsiflorus*, these stages were respectively about 32 and 42 days long.

These differences in the visibility of sex characteristics between males and females are statistically significant for both species. Generally, the female plants are recognizable for remarkably longer periods, however, this difference between sexes is more pronounced in the case of *R. acetosa*. Furthermore, in this species, durations of sex characteristics' visibility are less dispersed between individuals and the female characteristics are significantly longer distinguishable than in *R. thyrsiflorus* (Table 2, Fig. 2). On the contrary, male plants of *R. acetosa* seem to have shorter recognizability time, but the difference between species is

**Table 3  The comparisons of the sex characteristics visibility between male and female mature plants of *R. acetosa* and *R. thyrsiflorus* and between species for both species separately, for first flowering.** M, male; F, female; mean 1, mean 2, means of sex characteristics visibility in 1 and 2 sets (species and sex) respectively; difference, the difference between mean 1 and mean 2. CI-, CI+, lower and upper values of confidence intervals for the difference between mean 1 and mean 2. Asterisks (*) indicate statistically important differences between means with respect to Benjamini & Hochberg correction (*Benjamini & Hochberg, 1995*).

| Species 1 | Species 2 | Sex 1 | Sex 2 | Mean 1 | Mean 2 | Difference | *p* | CI- | CI+ |
|---|---|---|---|---|---|---|---|---|---|
| *R. acetosa* | *R. acetosa* | M | F | 26.76 | 49.18 | −22.41 | <0.01* | −24.28 | −20.55 |
| *R. thyrsiflorus* | *R. thyrsiflorus* | M | F | 31.56 | 42.06 | −10.5 | <0.01* | −16.37 | −4.64 |
| *R. acetosa* | *R. thyrsiflorus* | M | M | 26.76 | 31.56 | −4.79 | 0.08 | −10.3 | 0.72 |
| *R. acetosa* | *R. thyrsiflorus* | F | F | 49.18 | 42.06 | 7.12 | <0.01* | 3.97 | 10.27 |

shorter than in the case of female plants, and not statistically significant. Regarding female plants, it is worth noting that all female individuals with fruits were visible as long as there was no manual mowing/cutting down of the sprouts, so the difference between the species in this case is a consequence of the later onset of flowering of *R. thyrsiflorus*. Figure 3 shows the time of recognizability of every plant in the field studied.

### Sex characteristics' visibility in time

The changes in male individuals' frequency of *Rumex acetosa* and *R. thyrsiflorus* is displayed in Fig. 4. The plot illustrates evolving proportions of recognizable male individuals during the season until plant cutting, for both species separately and when treated as one species.

In the case of the frequency of males' changes, the beginning of the season is dramatically different. For *R. acetosa* (Fig. 4A) in the first days of observations, only males are recognizable, contrary, in *R. thyrsiflorus* (Fig. 4B), this phenomenon is not present. Later, during the following weeks, the differences between sexes are not statistically significant; however, after this period, females dominate, and the frequency of males is statistically significantly different from 0.5. Interestingly, the changes in sex proportions seem more dramatic in the case of *R. acetosa* than in *R. thyrsiflorus*. Moreover, only in the former species is there a period when male individuals overwhelm female ones (Figs. 4A, 4B).

Figure 4C shows the sex proportions in case these species had been treated as one species. A joint analysis of the two species was undertaken since, although they typically inhabit different environments, instances of co-presence in the same habitat do occur. Given that their flowering periods may overlap and their morphological similarities are considerable—in the past *R. thyrsiflorus* was regarded by some researchers as a subspecies of *R. acetosa* (*R. acetosa* subsp. *thyrsiflorus* Fingerh.) (*Čelakovský, 1887*; *Świetlińska, Łotocka-Jakubowska & Żuk, 1970*)—there exists a risk of conflating the two as a single population. The shape of the curve is mostly intermediate between the curves for both species; however, the beginning of the season is dominated by males, as in the case of *R. acetosa*.

## DISCUSSION

The research presented here was carried out on individuals of *R. acetosa* and *R. thyrsiflorus* grown in an experimental plot under partially controlled conditions. This allows tracing the process of flowering and (in female plants) fruiting at the level of individual plants,

tested populations, and sexes. Such observations, which seem straightforward, actually involve several methodological hurdles.

The first problem concerns which plant features or developmental stages should be considered when recognizing the sex of a plant in the field. Non-flowering and non-fruiting plants are generally excluded from data collection based on morphological sex characteristics. Only flowering plants are eligible for sex identification. However, two main questions arise and need to be addressed. First, should only plants with open flowers be counted, or should those with still-closed but identifiable sex-determination buds also be included? Second, should fruits be treated as a characteristic of sex identification? Female plants carrying fruits remain identifiable for a much longer period than male individuals, which lack easily detectable signs of their sex after flowering ends. Consequently, if fruits are counted as a sex identification criterion, the results may be biased toward females, especially during the late season.

Another challenge in observing plants with inflorescences containing a large number of flowers was their categorization. This difficulty arose because, at any given time, flowers at different stages of development could be present on the same inflorescence, ranging from closed buds to flowers producing pollen or beginning seed formation. Since it was impossible to examine each flower individually across so many inflorescences and plants, a consistent rule was adopted and adhered to throughout the study. Plants were always assigned to the category corresponding to the latest stage of flower development observed on a given inflorescence. The initial, pre-reproductive phase was defined as the vegetative stage, encompassing any plant that exclusively exhibited leaf growth without visible flower buds. The onset of the reproductive cycle was marked by the flowering stage, which commenced with the appearance of the first closed flower buds and continued through the period of open flowers. Following this, sex-specific stages were identified: for male plants, the pollination phase began with the dehiscence of the first visible pollen release. For female plants, the seed production phase was initiated by the first recognizable fruit. Finally, the post-reproductive senescence stage was identified by the visible degradation of plant tissues, such as leaf chlorosis (yellowing), after the completion of pollination for males or seed maturation for females.

Another key aspect of the study was survival across early developmental stages. The only period with noticeable mortality appears to be at the early stages of plant development, as many of the seeds did not germinate, and numerous small plants died. The germination tests for these sets of seeds were not performed; however, our observations in other experiments indicate that germination rates in optimal conditions are much higher and may reach almost 100% (*e.g.*, 97/100). Therefore, conditions in the experiment described above were probably not optimal. This seems to impact mainly female plants, especially of *R. acetosa*, but the differences in sex proportions between seeds and mature plants are marginal (1–3%) and not statistically significant, so it should not impact overall conclusions. However, it is worth noting that the number of mature plants counted was rather low, which impacts the statistical significance of comparisons. All seedlings that developed enough to be transplanted into the soil survived the winter and re-emerged in the spring with the beginning of the new growing season. Survival rates may have been
influenced by good growing conditions, such as poor competition, lack of competing plant species and herbivores, "on demand" water supply, *etc*. It is possible that under natural or semi-natural conditions, mortality would be much higher, influencing sex bias. In natural conditions, there are several factors impacting survival rates, which may eliminate mostly male plants (in case of the studied species), especially when the population is observed over several years, as mentioned, *e.g.*, by *Rychlewski & Zarzycki (1973)* and *Rychlewski & Zarzycki (1986)*. Our experiment was performed in conditions close to optimal for plant survival, without selection of weaker plants, at least after the earliest stages of development. However, the aim of the study was not to simulate the natural processes of one sex elimination but rather only the influence of flowering and fruiting periods on the detectable (by morphological characteristics) sex proportions in the population in subsequent days of the season (*Korpelainen, 1992*; *Korpelainen, 2002*; *Stehlik, Friedman & Barrett, 2008*). Such results can give us a clue as to how the time of surveillance can affect the observable sex ratio in the population.

The sex ratio in seeds in our experiment was shifted toward female dominance for both species. This effect was stronger in the case of *R. thyrsiflorus* and was only statistically significant there. However, it should be noted that this difference between species might result only from the seed origin. The literature shows that seed sex proportions of both species vary in different populations (51% to 59.7% for *R. thyrsiflorus* and 42.3% to 61.8% for *R. acetosa*), and at least in some cases, female bias may be stronger in *R. acetosa*. (*Zarzycki & Rychlewski, 1972*; *Rychlewski & Zarzycki, 1981*; *Rychlewski & Zarzycki, 1986*; *Korpelainen, 2002*; *Kwolek & Joachimiak, 2011*). However, the question of why there is primary female bias at all still needs an explanation. One of the possible explanations is the detrimental effects of the Y chromosome. During the evolution of plant sex chromosomes, their DNA content increases, and Y chromosomes become rich in repetitive sequences but contain relatively few genes, leading to their rapid degeneration. This degeneration, driven by suppressed recombination and the accumulation of deleterious mutations reduces gene dosage in the heterogametic sex and may require compensatory mechanisms. In species such as *R. acetosa* and *R. thyrsiflorus*, Y chromosome degeneration is thought to slow pollen germination, resulting in female-biased sex ratios (*Rychlewski & Zarzycki, 1986*; *Charlesworth & Guttman, 1999*; *Ainsworth, 2000*; *Shibata, Hizume & Kuroki, 2000*; *Stehlik & Barrett, 2005*). Other possible explanations of female bias found in the literature include certation (selective fertilization resulting from pollen tube competition) and maternal tissue support of pollen tube growth of female pollen tubes (*Correns, 1917*; *Correns, 1922*; *Correns, 1928*; *De Jong & Klinkhamer, 2002*; *Stehlik, Friedman & Barrett, 2008*) and sex ratio distorter/restorer systems (*Taylor, 1994*).

The evolution of observed sex ratios in species studied (Fig. 2) shows that the time of observations may have a significant influence on the results. It is especially visible at the beginning period of flowering in the case of the *R. acetosa*, when recognizable males dominate, and in the last period of observations, when male flowers gradually (in *R. thyrsiflorus*) or more rapidly (*R. acetosa*) disappear, but female flowers and especially fruits are still visible.

The phenomenon of earlier flowering in male plants, observed in the latter species, prompts an inquiry into its underlying causes. It may be hypothesized that evolutionary pressures would favor synchronized flowering between both sexes. However, it is important to consider whether, in certain scenarios, earlier male flowering could actually enhance successful pollination—for example, by increasing the likelihood of pollinating unfertilized pistils. Conversely, female plants do not appear to derive significant benefits from early flowering. Instead, they may invest in a longer flowering duration, which could lead to an increased number of seeds produced and greater genetic variability in their offspring, owing to the reception of pollen from a wider range of male plants.

It should be mentioned that in *R. acetosa*, during a relatively short period, the observed sex ratio does not differ significantly from the true sex proportion, regarded as the known proportion of male to female plants (the proportion of all observable males and females counted during the whole growing season). The period when the frequency of observable males was close to the real frequency of males in the population and when the difference between observable and real frequencies was not significant statistically, excluding the situations of too few (below 20) flowering plants, lasted about three weeks (23 days) for *R. acetosa* and more than 4 weeks (30 days) for *R. thyrsiflorus*. The presented plots (Figs. 2 and 4) may suggest when the best periods for sex ratio observations in the field are. But these data are potentially misleading. According to the literature, a distinguishing feature of the two species is their flowering period: *R. acetosa* typically flowers from May to June, while *R. thyrsiflorus* flowers from June to August (*Świetlińska, 1963*). But in our experimental field, these periods were significantly shifted toward the beginning of the season. This is especially visible in *R. thyrsiflorus*. The time of flowering started at the beginning of May, so about a month earlier. Literature data indicate that this species flowers until August, but in our experiment, even before the end of June, a noticeable deviation of the sex ratio from the actual one towards the female was observed. The shift in flowering time may be caused by various factors. The first one is climate change, which may influence the start of vegetation (*Geissler, Davidson & Niesenbaum, 2023*; *Pareja-Bonilla et al., 2025*). It is also possible that the specific genotype of plants used in the experiment (seeds of *R. thyrsiflorus* were commercially purchased) influenced the time of flowering.

## CONCLUSIONS

This study demonstrates that observable sex ratios in populations of *Rumex acetosa* and *R. thyrsiflorus* are dynamic rather than static, exhibiting significant fluctuations throughout the growing season as a consequence of differences in developmental timing and the duration of morphological recognizability between sexes. Our findings indicate that the interval during which field-observed sex ratios accurately represent the actual population structure is both limited and highly contingent upon the timing of observations. In *R. acetosa*, the earlier emergence of male individuals can result in a transient, but misleading, male-biased sex ratio during early-season surveys. Conversely, in both species, female plants remain morphologically identifiable for substantially longer periods, primarily due to the prolonged presence of fruits, thereby increasing the likelihood of a pronounced female bias in late-season assessments.

These results underscore the necessity for methodological rigor and temporal replication in demographic studies of dioecious plants, particularly in taxa exhibiting sex ratio biases. Reliance on single or infrequent observations may yield inaccurate or misleading conclusions regarding population sex structure, especially in species characterized by pronounced differences in the flowering of male and female individuals. Our work underscores the importance of carefully designed and implemented standardized, season-long monitoring protocols to ensure accurate assessment of sex ratios and to enhance understanding of the ecological and evolutionary dynamics governing dioecious plant populations. Overall, this research provides valuable methodological insights and a framework for future investigations into the population biology of dioecious, sex-biased species.

## ACKNOWLEDGEMENTS

Dr Pawełek thanks Dr. Tomasz Suchan for providing valuable comments and suggestions that helped improve the clarity and quality of the work.

### Funding

This work was supported by the W. Szafer Institute of Botany, Polish Academy of Sciences (the statutory funds), and Jagiellonian University, Faculty of Biology, Institute of Botany (No. K/DSC/003934, No. K/DSC/005523). The funders had no role in study design, data collection and analysis, decision to publish, or preparation of the manuscript.

### Grant Disclosures

The following grant information was disclosed by the authors:
W. Szafer Institute of Botany, Polish Academy of Sciences (the statutory funds).
Jagiellonian University, Faculty of Biology, Institute of Botany: No. K/DSC/003934, No. K/DSC/005523.

### Competing Interests

The authors declare there are no competing interests.

### Author Contributions

- Barbara Pawełek conceived and designed the experiments, performed the experiments, analyzed the data, prepared figures and/or tables, authored or reviewed drafts of the article, data curation, funding acquisition, project administration, resources, and approved the final draft.
- Dagmara Kwolek conceived and designed the experiments, analyzed the data, authored or reviewed drafts of the article, and approved the final draft.
- Grzegorz Góralski conceived and designed the experiments, analyzed the data, prepared figures and/or tables, authored or reviewed drafts of the article, and approved the final draft.

## Data Availability

Raw data and code are available in the Supplemental Files.

## Supplemental Information

Supplemental information for this article can be found online at http://dx.doi.org/10.7717/peerj.20391#supplemental-information.

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
