# Peer review of "Sex ratios in flux: seasonal dynamics and methodological insights in Rumex species"

_PeerJ, doi:10.7717/peerj.20391_

## Round 0.1 · original submission · Major Revisions

· Academic Editor

Major Revisions

Thank you very much for your manuscript titled “Sex ratios in flux: seasonal dynamics and methodological insights in Rumex species” that you sent to PeerJ.
This study presents very valuable and relevant information on the dynamics of the sex ratio in two Rumex species throughout an entire growing season under semi-controlled conditions.

As you will see below, comments from referee 2 suggest a minor revision while reviewers 1 and 3 suggest a major revision before your paper can be published. Given this, I would like to see a major revision dealing with the comments. Their comments should provide a clear idea for you to review, hopefully improving the clarity and rigor of the presentation of your work. I will be happy to accept your article pending further revisions, detailed by the referees.

Reviewer 1 suggests improving the presentation of the results, the discussion, and the conclusions. He also raises some questions about possible alternative ecological and evolutionary interpretations of the results.

Reviewer 2 points out some points that need clarification regarding the methods and results.

Reviewer 3 believes that the introduction and presentation of the results and conclusions need to be substantially improved. He also has some concerns about the origin of the seeds, their viability, germination, and possible biases in the environmental conditions of the crops. He also questions the resolution of plant sex determination.

Please note that we consider these revisions to be important and your revised manuscript will likely need to be revised again.

Reviewer 1 ·

Basic reporting

The study presents an interesting investigation into population-level phenomena in Rumex acetosa and R. thyrsiflorus, revealing a notable female-biased sex ratio in both species, more pronounced in R. thyrsiflorus. The language is mostly clear, with the exception of places I comment on later. The list of references is sufficient, though it might be slightly extended based on the revision. However, the manuscript currently falls short in its discussion, where the implications of these findings are not sufficiently contextualized or integrated with existing knowledge. Instead, the authors focus heavily on methodological challenges, which could be shortened to allow more space for interpreting the results.
Therefore, I recommend a substantial revision of the discussion section. Based on these revisions, minor improvements to the conclusion should also be considered before acceptance.

Major Concerns
• The authors should further elaborate on the observed female-biased sex ratio in R. thyrsiflorus, as summarized in Table 1. Is there a possible biological explanation, particularly in light of what is stated in the Introduction regarding bias in both seeds and mature plants? For instance, could this be linked to the detrimental effects of the Y chromosome?
• Similarly, Figure 2 should be discussed in greater detail. Why do males appear to flower earlier than females? Could this be attributed to co-evolution between the sexes, possibly involving genes such as FLC, which may be differentially regulated in males and females? Might this reflect an evolutionary pressure for male germlines to initiate flowering earlier to maximize pollination opportunities, while females benefit from prolonged flowering for higher pollination success? These ideas could be integrated with the concepts already mentioned in the Introduction.
• Throughout the manuscript, the terms “sex features” and “sex characteristics” are used interchangeably. It would improve clarity to use a consistent term throughout, or alternatively, provide a clear definition of each term upon first use. Additionally, I recommend including a figure depicting R. acetosa and R. thyrsiflorus, highlighting key floral characteristics and organ structures, with appropriate labels (e.g., "male floral organs," "female floral organs") to help readers better understand the text.
• Lines 140–155 could serve as a natural transition into the Results section. Consider using this part to introduce how the current study addresses the previously mentioned challenges.
• After reading the manuscript, I had the following question: Do the authors expect that sex bias in seeds might shift during the vegetation period? For example, could earlier-produced seeds exhibit a slightly different sex ratio than those produced later in the season? In this context, might female plants produce fewer seeds during the hotter mid-season, with seed production increasing again in the cooler late season?

Minor Concerns
• Lines 17–18: I recommend shortening this sentence. As the number of plants with characterized sex chromosomes is continually increasing, the current figure will soon be outdated. Moreover, this information is already mentioned in the Introduction and may be unnecessary in the Abstract.
• Line 101: Consider opening this section with a brief explanation of the biological significance of the Rumex genus, including its relevance in plant evolutionary studies, ecology, or chromosome biology.
• Line 346: Please clarify what is meant by “cutting the plants.” If the plants were terminated or removed from the experiment, state this explicitly. Similar clarifications are needed elsewhere.
• Figure 1: The color used for the “winter” category should be adjusted to improve visibility.
• Figure legends: Each figure legend should be self-explanatory. Please ensure that all figure captions include a brief but informative biological context to aid the reader's interpretation.

Experimental design

Research questions are defined meaningfully. Methods are described sufficiently and should be easily replicated. Nevertheless, it would be helpful for readers if the authors could provide a schematic diagram (in the supplementary material) outlining the experimental timeline and sampling procedure. This should summarize the data collection process, relevant environmental variables, and other experimental factors described in the Materials and Methods section.

Validity of the findings

Based on my previous comments, the conclusion and discussion should be revisited.

Reviewer 2 ·

Basic reporting

The authors conducted a multi-season, multi-method, and well-designed investigation into sex ratio variation in Rumex species. This study provides a renewed perspective on the sex ratio in dioecious plants and offers insights into why statistics from the earlier studies may be unreliable. The findings presented by the authors are likely to guide subsequent research in this area. However, there are some unclear points.

Experimental design

-

Validity of the findings

-

Additional comments

1. Were the seeds used for sex identification taken from the same pool as those that were sown for the germination experiment?

2. The authors used PCR for species confirmation of male seeds. However, how did the seed sex be identified? Please provide the PCR results.

3. In the Materials & Methods section, it is stated that molecular methods were used to identify the sex of “individuals”. Does this refer only to seeds, or were any mature plants also included in the molecular analysis as a reference?

4. “The developmental dynamics of R. acetosa and R. thyrsiflorus were monitored throughout the entire growing season of 2021.” Specific dates? Will this change in different years?

5. Figure 4 shows the sex ratio of flowering individuals or all individuals?

6. Lines 259 – 261 contain two sentences that are repetitive.

7. In line 379, “subsp.” and the authority in the scientific name should not be in italics.

8. The authors said, “this research provides valuable methodological insights and a framework for future investigations into the population biology of dioecious, sex-biased species”. One should investigate the sex ratio by “Molecular sex determination”? Right?

9. Please provide photos related to the experiment, such as cultivation equipment, seeds, plants, etc.

·

Basic reporting

This manuscript presents an interesting investigation into sex expression and phenology in two Rumex species and certainly should be published. However, several areas of concern need to be addressed to improve the rigour and clarity of the manuscript before it can be published.

Experimental design

• Seed Source and Germination: The germination rates reported are extremely low, raising red flags about seed viability and experimental consistency. It appears that no viability testing was conducted before sowing. Additionally, there is no detailed description of the germination or growing conditions (e.g., soil type, pH, fertilisation, light exposure), which are critical for reproducibility and interpretation of the results.
• Experimental Design and Conditions: Growing plants “on a patio” does not constitute controlled conditions. This, along with the apparent plant dieback after flowering, suggests suboptimal or inconsistent cultivation practices. These environmental factors may have introduced substantial bias into the observed phenological differences and sex ratios.
• Sampling Strategy and Consistency: The source of the seeds is not consistently reported, with R. acetosa collected from the wild and R. thyrsiflorus obtained commercially. This inconsistency could introduce bias, particularly if the commercial seeds were selectively bred. Voucher specimens and locality data should be included for wild-collected material to ensure transparency and traceability.
• Sex Determination and Morphological Identification: The criteria used to differentiate male and female plants are not described in sufficient detail. This is particularly problematic given that early sex identification is a key outcome of the study. The results could have been strengthened by incorporating molecular data (e.g., PCR, which was suggested), especially since the authors mention genetic differences early in the manuscript.

Validity of the findings

• Data Presentation: Several data summaries are vague or qualitative (e.g., “almost all male plants began flowering earlier”). This is not appropriate for a scientific paper and needs to be supported by actual numbers, ranges, and statistical comparisons. Figures and tables should be clearly referenced and meaningfully interpreted in the text.
• Literature Context and Framing: The introduction lacks balance, focusing on angiosperms while omitting key examples from gymnosperms and animals, despite these being relevant to dioecy and sex determination. Recent advances in whole-genome sequencing and sex chromosome identification should also be integrated into the discussion.
• Clarity and Internal Consistency: There are contradictions between the introduction and the results/discussion sections regarding genetic differentiation between the species. These inconsistencies need to be clarified to avoid undermining the validity of the conclusions.

Additional comments

Page-specific comments

50: I'm unsure why only angiosperms are included here, since the broader context considers animals too. I believe gymnosperms should also be included because a larger percentage of them are actually dioecious.

54: Whole genome sequencing is also adding significantly to finding genes that relate to sex determination on chromosomes. This needs to be considered, and not that of chromosome detection.

101: “Despite the general trend of male-biased sex ratios in dioecious plants, female-biased sex ratios are observed in natural populations of R. acetosa L. (common sorrel) and R. thyrsiflorus Fingerh. (thyrse sorrel), both members of the family Polygonaceae, subgenus Acetosa. This sentence seems to repeat the example provided above and seems superfluous.

168: It would be good to know where the seeds were collected from in the wild and have this data in the table and associated with the field voucher specimen.

168: Why were seeds of R. thyrisflorus not collected from the wild? It seems inconsistent and biased. Do you know if the purchased seeds have been selected breed?

196 to 197: Please briefly describe these methods in short form, as it will greatly aid the reader.

226: These are very low germination rates. It seems like no seed viability testing was conducted beforehand, and the growing conditions were likely not correct. I feel you need to state potential reasons why this could be, as at present it is a red flag.

237 to 240: I feel these results could be quite biased, as the growing medium being used has not been specified, including the soil pH. Also, were these plants fertilised, and was a growing medium used that contained organic matter?

244 to 245: This data needs to be summarised better, and give an overview of the dates and number of days to maturity, and other data from the table. Why have a figure in the paper when the results are not being summarised or explained?

251 to 253: Give numbers here and provide the range of the measures. For example, what was the earliest-flowering female plant, and what was the latest? Provide the meaning. This may need to be done in a table and summarised. Stating that “Almost all male plants began flowering earlier than females” is not acceptable in any scientific paper.

309 to 314: I find it unusual that these perennial herbs are dying after flowering, as this indicates issues with the growing conditions. I think this needs to be addressed.

346: So why mention this if you did not test it, as it seems you are potentially biasing results by cutting the plants down, to stimulate an unnatural process. I think this needs to be clarified.

379 to 380: At the start of the paper, it was mentioned that there were genetic differences between these samples as the seeds of R. acetosa turned out to be R. thyrsiflorus. The comments here make me suspect this is not the case. Thus, I feel you need to go into more detail here.

388: Again, you need to provide the reader with these conditions.

395 to 396: This is a question that needs to be addressed within the paper directly and not just in the discussion. You should be stating more clearly what you are determined to be as an open flower, and also what state they were recorded in.

398 to 399: To me, this indicates that the methods of differentiating a male from a female should be quite clear using a hand lens. The stamen of the males should still remain intact.

447 to 448: This could indicate some degree of hybridisation, as seeds were purchased and likely of unknown origin. Also, phenological shifts could be related to growing conditions.

454 to 456: “placed on a patio.” This is not controlled conditions for this type of experiment and would likely influence the results.

---

## Round 0.2 · Minor Revisions

· Academic Editor

Minor Revisions

After reviewing this revised version of your manuscript, I see that the main comments suggested by the reviewers have been included. However, there are still some details that need to be clarified before having a final version that can be published.

It is necessary to clarify some contextual and methodological points as pointed out by reviewer 3.

Reviewer 1 ·

Basic reporting

Based on the current version of the manuscript, I do not have further comments, except for the newly added text regarding lines 61–67. The phrasing in lines 63–65 seems somewhat unclear, particularly the part: “also for the study of the functions of sex-related genes found within them and other features of the mentioned chromosomes.” The authors could consider rewording this more explicitly to clarify what NGS techniques allow researchers to investigate.

For line 66, I recommend placing the reference to Akagi et al. (2018) directly after the first mention of kiwifruit, to avoid biasing the reader. Regarding the references to Michalovová and Jesionek, it is correct that they apply NGS and/or RNA-seq profiling. However, if the journal’s reference limit allows, I suggest adding a more recent citation on the Rumex genus that integrates both long-read sequencing and RNA-seq data, for example:
https://academic.oup.com/evlett/article/9/2/221/7905796?login=true
https://academic.oup.com/mbe/article/41/4/msae074/7644656?login=true

Experimental design

No comment. Based on my previous comments, the authors already included new figure(s) and text.

Validity of the findings

No comment.

Reviewer 2 ·

Basic reporting

The authors have addressed all the questions I raised. Thanks.

Experimental design

No.

Validity of the findings

no comment

·

Basic reporting

The manuscript requires clearer contextual and methodological detail throughout. The authors should include full locality information, remove unnecessary author references, and clarify several ambiguous statements. Greater precision is needed when describing germination timing, flowering overlap, and the rationale for not conducting germination tests. The discussion should be restructured to separate and define the two main questions more clearly, and the statement on sex ratios should be revised for clarity. Finally, the lack of temperature data represents a methodological limitation that needs to be addressed carefully to avoid appearing as poor experimental design.

Experimental design

185 to 186: Please just state the locality, state and country, not just the coordinates. This question was previously asked and has not been addressed.

190 to 191: No need to state the main author here. Just state the research collection.

291: You are stating that one species began to germinate sooner than the other. You need to specify the timeframe here and the number/frequency of individuals in which this occurred.

366 to 368: If the female were over 10 days longer, was there much overlap between males and females? I assume that not all males that produce flowers are the same, and there should be an overlap period to ensure maximum fertilisation chances.

Validity of the findings

442 to 443: This point needs to be explained better, as it is not easy to follow.

446 to 451: Two questions are being presented here and need to be defined better, e.g. “However, there is a question” and “There is also another question”; it would be better to break this down further and state there are two main questions that arise that need to be addressed within the discussion.

471 to 472: It is important to state why there were no germination tests completed and to have a compelling reason. Because, at present, this comes across as poor project planning.

529 to 530: What do you mean by this statement, “the observed sex ratio does not differ significantly from the real one”? It is unclear, and the statement seems incomplete.

546 to 547: The lack of temperature recording here is a significant issue with the approach. I feel you need to be very careful how you mention this.

---

## Round 0.3 · accepted · Accept

· Academic Editor

Accept

After reviewing this revised version of your manuscript, I see that the main comments suggested by the reviewers have been included, while the suggestions not considered are justified in detail. Therefore, I am satisfied with the current version and consider it ready for publication.

For instance, the Section Editor noted:

> My main concern is that the abstract does not convey any of the results of this experimental study. I was expecting the abstract to include some of the key conclusions summarized on lines 564-573. I think some introductory/background material could be removed from the Abstract to make room to add some key results/conclusions.
>
> Regarding Figure 2, the Legend refers to "winter" but I did not find the color corresponding to "winter" anywhere on Figure 2.